# DSPO: Direct Semantic Preference Optimization for Real-World Image Super-Resolution

## Abstract

Recent advances in diffusion models have improved Real-World Image Super-Resolution (Real-ISR), but lack human feedback integration, risking misalignment with human preference and potentially leading to artifacts, hallucinations, and harmful content generation. To this end, we are the first to introduce human preference alignment into Real-ISR, a technique that has been successfully applied in Large Language Models and Text-to-Image tasks to effectively enhance the alignment of generated outputs with human preferences. Specifically, we introduce Direct Preference Optimization (DPO) into Real-ISR to achieve alignment, where DPO serves as a general alignment technique that directly optimizes from the human preference. Nevertheless, the pixel-level reconstruction objectives of Real-ISR are difficult to reconcile with the image-level preferences of DPO, which can lead to the DPO being overly sensitive to local anomalies, leading to reduced generation quality. To resolve this challenge, we propose Direct Semantic Preference Optimization (DSPO) to align instance-level human preferences by incorporating semantic guidance, which consists of two strategies: (a) semantic instance alignment strategy, implementing instance-level alignment to ensure fine-grained perceptual consistency, and (b) user description feedback strategy, mitigating hallucinations through injecting user semantic textual feedback on instance images as prompt guidance. Our method surpasses both Real-ISR and preference alignment baselines, demonstrating superior performance. As a plug-and-play solution, DSPO performs consistently across one-step and multi-step SR frameworks, highlighting strong generalizability.

## 1 Introduction

Real-world Image Super-Resolution (Real-ISR) Chen et al. (2022); Zhang et al. (2023); Wang et al. (2024a) aims to reconstruct photo-realistic high-quality (HQ) images from low-quality (LQ) images with various degradations such as noise, blur, and low-resolution. Recently, diffusion models Ho et al. (2020); Dhariwal & Nichol (2021); Song et al. (2020) have made excellent progress in Real-ISR tasks Wu et al. (2025); Wang et al. (2024a); Yang et al. (2024); Lin et al. (2024); Wu et al. (2024); Yu et al. (2024), owing to their remarkable capability of generation. However, these models generally employ the supervised training paradigm that directly learns from paired LQ-HQ image datasets, omitting human feedback throughout the training cycle. Without human intervention, the optimization objectives of these models may misalign with human perceptual preferences, leading to potentially harmful content generation, hallucination phenomena, and visual artifacts.

Such misalignment between model outputs and human preferences also exists in other tasks. For instance, in the fields of Large Language Models (LLMs)Achiam et al. (2023); Touvron et al. (2023) and Text-to-Image (T2I) generationWallace et al. (2024); Li et al. (2025), human preference alignment techniques have been widely employed to mitigate misalignment issues by fine-tuning the pre-trained model through Reinforcement Learning from Human Feedback (RLHF) strategies. The classical RLHF paradigm (*e.g.*, PPO Schulman et al. (2017), DDPO Ho et al. (2020)) first trains a reward model on a fixed preference dataset, then optimizes the policy to maximize the predicted reward. However, relying on a reward model makes the process inherently complex and significantly increases computational overhead Rafailov et al. (2023). In contrast, Direct Preference Optimization (DPO) Rafailov et al. (2023) is proposed to directly optimize the policy model based on human

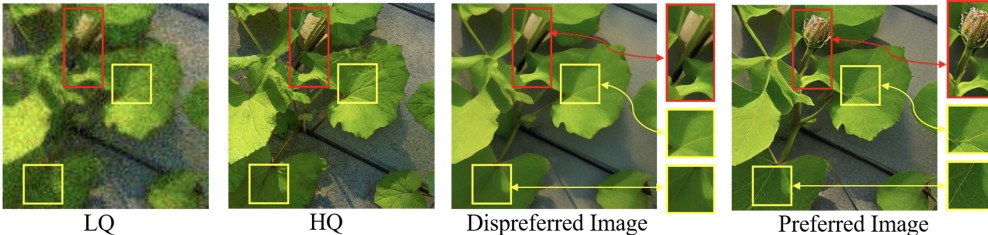

| LQ | HQ | Dispreferred Image | Preferred Image |

Figure 1: The dilemma between image-level preferences in DPO and pixel-level reconstruction objectives in Real-ISR: The preferred image, selected by the 'winner of overall image visual pleasure' rule, appears sharper in the yellow bbox and other areas but shows local hallucinations in the red bbox, where the dispreferred image performs better.

preference data without the reward model and demonstrate excellent performance on generative tasks Rafailov et al. (2023); Wallace et al. (2024).

Despite success in LLM and T2I, human preference alignment (*e.g.* DPO) remains unexplored in Real-ISR. Therefore, we introduce human preference alignment into Real-ISR for the first time through DPO. However, directly applying DPO to Real-ISR results in performance degradation due to the inherent dilemma between image-level preferences of existing DPO and the pixel-level reconstruction objectives of Real-ISR tasks. Specifically, as illustrated in Fig. 1, image-level preferences may lead to artifacts or hallucinations in preferred images within local regions, especially in high-frequency texture and complex regions. Such conflicts may lead to the models being overly sensitive to local anomalies, exhibiting fluctuations and ambiguity during training, which ultimately affects the quality of the generated performance.

To address this problem, we propose Direct Semantic Preference Optimization (DSPO), which deeply aligns instance-level human preferences by incorporating semantic guidance. Specifically, we propose the semantic instance alignment strategy that conducts human preference alignment at the instance level to achieve finer-grained alignment. An instance extractor is employed to extract individual instances from SR outputs, and then preferred and dispreferred instance-level cases are selected, followed by instance-level preference alignment. Additionally, to further mitigate the hallucination phenomenon, we propose the user description feedback strategy: we incorporate users' semantic textual feedback on instance-level images and select hallucination semantic information texts as prompt injection.

Our contributions are as follows: (1) We pioneer introducing human preference alignment into Real-ISR, establishing the first methodological approach to incorporate human preference alignment in this field. (2) We propose DSPO, which achieves instance-level human preference alignment and significantly suppresses artifacts and hallucination phenomena. (3) Our method outperforms both existing Real-ISR approaches and preference alignment baselines, demonstrating its superior performance. (4) As a plug-and-play solution, DSPO achieves consistent effectiveness across both one-step and multi-step SR frameworks, highlighting its strong generalizability.

## 2 RELATED WORK

**Generative SR Models** Traditional super-resolution (SR) methods based on Convolutional Neural Networks (CNNs) Dong et al. (2014) and Generative Adversarial Networks (GANs) Ledig et al. (2017) focus on pixel fidelity and perceptual quality, while diffusion models achieve superior SR performance with stronger generative capabilities Saharia et al. (2022). Diffusion models have become central to Real-ISR by restoring high-quality images through stepwise denoising. While early methods based on Denoising Diffusion Probabilistic Models (DDPM) Kawar et al. (2022); Song et al. (2020) struggle with complex degradations, recent approaches Yu et al. (2024); Yang et al. (2024); Wang et al. (2024a); Wu et al. (2024) address these challenges. StableSR Wang et al. (2024a) integrates a temporal-aware encoder to improve recovery quality, and SeeSR Wu et al. (2024) leverages text guidance to improve semantic consistency and detail. One-step diffusion methods further accelerate inference. OSEDiff Wu et al. (2025) employs Variational Score Distillation (VSD) to boost efficiency and performance.

**Human Preference Alignment in LLMs**  To align with human preferences, LLMs are typically first supervised fine-tuned (SFT) and then optimized via RLHF Ouyang et al. (2022). Traditional RLHF methods, such as Proximal Policy Optimization (PPO) Schulman et al. (2017), rely on reward models to guide policy learning but struggle with challenges to train a reward model when the reward signal is unclear, suffering from high computational costs and training instability. To overcome the limitations, Direct Preference Optimization (DPO) serves as an alternative method, allowing LLMs to optimize directly based on pairwise preference data without training a reward model Rafailov et al. (2023). DPO has low computational overhead and stable optimization, demonstrating superior performance on open-source models like Llama 2 Bai et al. (2022). Compared to reward model-based methods, DPO is more efficient in optimizing LLM preferences, reducing training complexity while maintaining competitive performance Touvron et al. (2019).

**Human Preference Alignment in T2I**  Human preference alignment has emerged as a key direction for enhancing the subjective quality of T2I tasks. ImageReward Xu et al. (2023) trains reward models using human rating data to optimize generative preferences. However, this method is susceptible to bias and has limited generalization capabilities Bai et al. (2022). DDPO Ho et al. (2020) optimizes diffusion models within a small vocabulary range but struggles to adapt to complex text prompts, highlighting the limitations associated with reward model-based approaches. In contrast, Diffusion-DPO fine-tunes diffusion models directly based on human preference data without requiring an explicit reward model Wallace et al. (2024), enhancing the generation quality for open vocabulary without increasing inference costs, thereby aligning T2I tasks more closely with human aesthetics and semantic consistency.

# 3 PRELIMINARIES

**DPO in LLM Tasks**  Direct Preference Optimization (DPO) Rafailov et al. (2023) is a preference alignment method that does not require training a reward model, and is applicable for optimizing LLM. DPO optimizes the generation probabilities of paired preference data $(x_w, x_l)$ such that the preferred sample $x_w$ has a higher probability than the non-preferred sample $x_l$. The DPO objective can be expressed as:

$$L = -\mathbb{E}_{(c,x_w,x_l)\sim D}\left[\log\sigma\left(\beta\log\frac{p_\theta(x_w|c)}{p_{\text{ref}}(x_w|c)} - \beta\log\frac{p_\theta(x_l|c)}{p_{\text{ref}}(x_l|c)}\right)\right],\tag{1}$$

where $p_\theta(x|c)$ and $p_{\text{ref}}(x|c)$ represent the probability distribution generated by the DPO-trained LLM and the reference (pre-trained) model, respectively. The function $\sigma(x)$ is the sigmoid function, and $\beta$ controls the regularization strength.

**Diffusion-DPO in T2I Tasks**  In Text-to-Image (T2I) tasks, the sampling process of diffusion models is executed step-by-step, where the objective is not to directly optimize the final generated image but to influence the denoising process at each time step $t$, enhancing the likelihood of recovering preferred samples. Thus, the DPO objective can be extended to the diffusion process, optimizing the denoising probability distribution at each time step $t$:

$$L = -\mathbb{E}_{(c,x_w,x_l)\sim D,t\sim U(0,T)}\log\sigma\Bigg[\beta\mathbb{E}_{x_{1:T}^w\sim p_\theta(x_{1:T}^w|x_0^w)}$$

$$\log\frac{p_\theta(x_0^w|x_{1:T}^w)}{p_{\text{ref}}(x_0^w|x_{1:T}^w)} - \beta\mathbb{E}_{x_{1:T}^l\sim p_\theta(x_{1:T}^l|x_0^l)}\log\frac{p_\theta(x_0^l|x_{1:T}^l)}{p_{\text{ref}}(x_0^l|x_{1:T}^l)}\Bigg],\tag{2}$$

Here, the condition text is compactness. $x_{0:T}$ denotes the complete diffusion path, $p_\theta(x_{1:T}|x_0)$ represents the diffusion process given the initial state $x_0$, $p_\theta(x_0|x_{1:T})$ is the denoising probability distribution after the given diffusion trajectory $x_{1:T}$, and $p_{\text{ref}}(x_0|x_{1:T})$ refers to the corresponding distribution of the reference model. By optimizing the log probability ratio throughout the diffusion process, DPO-T2I encourages the model to generate images that align more closely with human preferences, thereby enhancing the quality of alignment in T2I tasks.

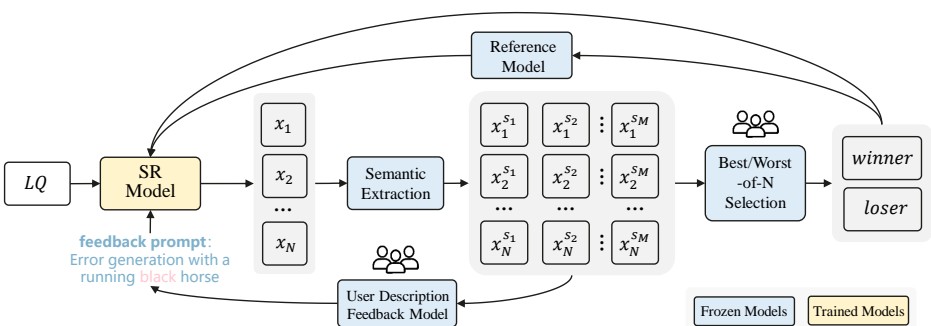

Figure 2: The overview of the proposed Direct Semantic Preference Optimization (DSPO) method.

## 4 METHODOLOGY

### 4.1 OVERVIEW

We propose Direct Semantic Preference Optimization (DSPO), which enhances instance-level human preference alignment by integrating semantic guidance. It is important to note that DSPO is designed as a fine-tuning approach for SR models. Therefore, DSPO is applied only during the training phase, while the inference stage remains identical to the pre-trained SR model to enable direct comparison. The overview of the proposed DSPO method is illustrated in Fig.2. Specifically, to achieve finer-grained alignment, we design the semantic instance alignment strategy (detailed introduced in Sec. 4.2): DSPO first generates a series of SR output with the same input LQ image using a pre-trained SR model under various settings and uses a semantic extraction model (*e.g.*, SAM Kirillov et al. (2023)) to extract instance-level semantics. By selecting the Best/Worst-of-N, it obtains the fine-grained winners and losers, which will align with SR models through DPO. Additionally, to mitigate hallucination in local regions, we propose a user description feedback strategy (detailed introduced in Sec. 4.3), which uses a Vision-Language Model(VLM) model to describe multiple instance-level samples and feeds semantically misaligned examples back to the prompt branch of the SR model by prompt injection.

### 4.2 SEMANTIC INSTANCE ALIGNMENT STRATEGY

We propose the semantic instance alignment strategy, which aligns human preferences at the instance level to achieve finer-grained alignment. As exhibited in Fig. 2, based on the pre-trained SR, we first generate a series of SR output with the same input LQ image using pre-trained SR model under various settings. For each SR output, we implement an semantic extraction model (*i.e.* Segment Anything (SAM) Kirillov et al. (2023)) to generate instances. Subsequently, we select instance-level preferred and dispreferred examples based on human evaluations and apply DPO at the instance level. The detailed implementation is as follows.

### 4.2.1 OPTIMIZATION OBJECTIVE

For an input low-resolution image $x_{\text{LQ}}$, we first generate a series of SR outputs $\{x_1, x_2, \ldots, x_N\}$. For each SR image $x_i$ generated from the pre-trained model, we generate $M$ semantic instance regions using SAM Kirillov et al. (2023): $S = \{s_m\}_{m=1}^{M}$, $\sum_{m=1}^{M} s_m = \mathbf{1}$, where each region $s_m$ represents to the mask of the $m$-th semantic instance, and $s_m \in \{0,1\}^{H \times W}$ is a binary mask matrix representing the spatial coverage of the semantic instance. Here, $M$ is not a fixed constant but depends on the number of instances automatically segmented by SAM, which varies from image to image. In the construction process of the optimization objective, different semantic instances have different contributions to optimization due to their varying size of areas. Therefore, the weight $w_m$ for optimization intensity of each semantic instance is defined as follows:

$$w_m = \frac{|s_m|}{\sum_{m=1}^{M} |s_m|},\tag{3}$$

where $|s_m|$ denotes the number of pixels in the semantic instance $s_m$.

After segment, each SR image $x_i$ can be divided into different instances $\{x_i^{s_1}, x_i^{s_2}, \ldots, x_i^{s_M}\}$. This allows us to obtain the different instance-level image within the same instance $s_m$ region across different images: $\{x_1^{s_m}, x_2^{s_m}, \ldots, x_N^{s_m}\}$. We then perform a human preference-based Best/Worst-of-N selection and obtain $x_w^{s_m}$ and $x_l^{s_m}$, respectively. The whole image $x_w$ of the best instance $x_w^{s_m}$ is defined as the preferred example, while the whole image $x_l$ of $x_l^{s_m}$ is dispreferred example. The final optimization objective is as follows:

$$\max_\theta \mathbb{E}_{(x_{\text{LQ}}, x_w, x_l) \sim D} \sum_{m=1}^{M} w_m \cdot \left[ \log \sigma \left( \beta \log \frac{p_\theta(x_w | x_{\text{LQ}}, s_m)}{p_{\text{ref}}(x_w | x_{\text{LQ}}, s_m)} - \beta \log \frac{p_\theta(x_l | x_{\text{LQ}}, s_m)}{p_{\text{ref}}(x_l | x_{\text{LQ}}, s_m)} \right) \right], \quad (4)$$

where $p_\theta(x | x_{\text{LQ}}, s_m)$ represents the probability of the target model generating an SR image $x$ within the semantic instance $s_m$, while $p_{\text{ref}}(x | x_{\text{LQ}}, s_m)$ represents the same for the reference (pre-trained) model.

### 4.2.2 LOSS FUNCTION

Referring to Diffusion-DPO Wallace et al. (2024), the optimization objective of DPO is further reformulated into a loss function grounded in noise prediction error within the framework of diffusion models, thereby ensuring that optimization is calculated progressively at each time step $t$ throughout the diffusion process. Therefore, our loss function can be expressed as follows:

$$L_{\text{SR}} = -\mathbb{E}_{(x_{\text{LQ}}, x_w, x_l) \sim D, t \sim U(0,T)} \left[ \sum_{m=1}^{M} w_m \log \sigma \left( -\beta T \times \left( L_{\text{diff}}^\theta(x_w, t, s_m | x_{\text{LQ}}) - \right. \right. \right.$$
$$\left. \left. \left. L_{\text{diff}}^{\text{ref}}(x_w, t, s_m | x_{\text{LQ}}) - \left( L_{\text{diff}}^\theta(x_l, t, s_m | x_{\text{LQ}}) - L_{\text{diff}}^{\text{ref}}(x_l, t, s_m | x_{\text{LQ}}) \right) \right) \right) \right], \quad (5)$$

where $L_{\text{diff}}(x, t, s_m | x_{\text{LQ}})$ represents the denoising error within the semantic instance $s_m$, serving as a stability constraint during training to reduce distribution shift. $L_{\text{diff}}^\theta$ and $L_{\text{diff}}^{\text{ref}}$ calculate the prediction error of the current model $\theta$ and reference model, respectively, and are defined as follows:

$$L_{\text{diff}}^{\text{ref}}(x, t, s_m | x_{\text{LQ}}) = \gamma(\lambda_t) ||s_m \cdot (\epsilon - \epsilon_{\text{ref}}(x, t | x_{\text{LQ}}))||^2, \quad (6)$$

$$L_{\text{diff}}^\theta(x, t, s_m | x_{\text{LQ}}) = \gamma(\lambda_t) ||s_m \cdot (\epsilon - \epsilon_\theta(x, t | x_{\text{LQ}}))||^2, \quad (7)$$

where $\epsilon_\theta(x, t | x_{\text{LQ}})$ denotes the noise predicted by the diffusion model at time step $t$, where $\epsilon$ represents the true noise. $\lambda_t$ represents the signal-to-noise ratio Kingma et al. (2021), and $\gamma(\lambda_t)$ is a predefined weighting function, often set as constant Ho et al. (2020); Song & Ermon (2019). This loss function enables optimization at the semantic instance level at each time step, enabling the SR model to align with human perception in finer-grained regions. Additionally, it leverages the reference model to prevent distribution drift. We compute the loss over the full image and apply it selectively via the segmentation mask $S_m$. This preserves global context, allowing the pretrained diffusion model to leverage global priors during DSPO optimization. In contrast, using segmented patches lacks global context. Moreover, varying segment sizes disrupt positional encoding, degrading spatial modeling and training stability.

### 4.3 USER DESCRIPTION FEEDBACK STRATEGY

We further propose the user description feedback strategy to relieve the hallucination problem of generative SR models. The SR outputs $x_i \in (x_1, x_2, \ldots, x_N)$ are segmented to extract individual semantic instances, which are then analyzed through the VLM for descriptive analysis. By employing human evaluations, we identify semantic instances that do not align with the input low-quality (LQ) content and gather their corresponding text descriptions to form error generation prompts. These prompts are utilized to constrain the generation direction during the sampling process of the diffusion model to avoid hallucination. This approach ensures that the diffusion model can minimize hallucination generation throughout the optimization process, enhancing semantic consistency in the super-resolution task.

### 4.4 DSPO OBJECTIVES

The final loss function can be represented as follows:

$$L_{\text{DSPO}} = \sum_m L(x_w, x_l | x_{\text{LQ}}, s_m, p_{\text{negative}}). \tag{8}$$

The DSPO strategy combines semantic instance alignment strategy and user description feedback strategy significantly enhancing the fine-grained recovery capability of SR models through optimization at the semantic instance level, while effectively mitigating the issue of hallucination generation using feedback description prompts. DSPO enhances the model's understanding of human preferences and improves its SR ability by addressing artifacts and hallucinations, especially in high-frequency texture and complex regions.

## 5 IMPLEMENTATION

### 5.1 TRAINING

#### 5.1.1 PRE-TRAINING CONFIGURATION

For pre-training, we evaluate DSPO across four distinct SR frameworks to assess its generalizability: OSEDiff, SD2.1[1], SeeSR, and AddSR Xie et al. (2024). Each framework is trained on the LSDIR dataset Li et al. (2023), with model inputs randomly cropped to $512 \times 512$. Low-quality (LQ) images are generated using the degradation pipeline from Real-ESRGAN Wang et al. (2021), resulting in 84991 LQ-HQ pairs. The SD2.1 using the Adam optimizer Loshchilov & Hutter (2017) for 150K iterations, with a batch size of 192 and a learning rate of $5 \times 10^{-5}$ to obtain the pre-trained muti-step SR model. During inference, we adopt spaced DDPM sampling Nichol & Dhariwal (2021) with 50 steps and set the cfg to 5.5. The pre-trained settings for OSEDiff, SeeSR, and AddSR remain consistent with their original settings in their paper. In the following experiments, we default to using OSEDiff for one-step frameworks and SD2.1 for multi-step frameworks unless otherwise specified.

**Dataset Preparation**   The sampling distribution between preference pairs should not vary too much Guo et al. (2024). Therefore, we obtain preference data candidates by adjusting the hyperparameters (e.g., step, cfg) of the pre-trained model instead of changing the SR model, thereby avoiding excessive sample deviation. Specifically, we apply different hyperparameter settings to a pre-trained model to infer multiple SR outputs for one LQ image. These outputs are evaluated using four representative SR metrics: PSNR Wang et al. (2004), SSIM Wang et al. (2004), NIQE Zhang et al. (2015), and CLIP-IQA Wang et al. (2023), which reflect pixel fidelity, structural consistency, perceptual quality, and semantic alignment, respectively. All scores are normalized, and the top four results with the highest average scores are selected as candidates. To retain the model's perception of negative samples, the default output of the pre-trained model is also included, replacing the lowest one if not already among the top four. After obtaining four different SR results, we use SAM Kirillov et al. (2023) to segment them into distinct instance regions. We then evaluate the same instance regions from the four results using two approaches: the human annotator method and the automatic scoring method, which are introduced as follows.

**Human Annotator Method**   We invite ten professionals specialized in low-level tasks to perform instance-level rankings, selecting preferred and dispreferred instance images. In addition, for the user description feedback strategy, we include the text results generated by Qwen2.5-VL-max Bai et al. (2025) below each image in the interactive interface. If the generated text does not align with the instance image, the user will select it as the feedback textual prompt. To reduce annotation costs, we select the first 500 images from LSDIR and annotate the top 5 largest segmented instance regions in each image, resulting in 2500 regions in total.

**Automatic Scoring Method**   To simulate human preference selection, we employ Qwen2.5-VL-max to compare SR results within each instance region. Specifically, four candidate images are input with a prompt asking the model to select the best and worst based on multiple aspects such as quality, realism, and consistency, forming a preference pair. Note that, unlike traditional IQA metrics that assess a single image from a single aspect, Qwen2.5-VL-max supports multi-image comparison and holistic evaluation across multiple dimensions, better aligning DSPO's pairwise

---

[1]https://huggingface.co/stabilityai/stable-diffusion-2-1-base

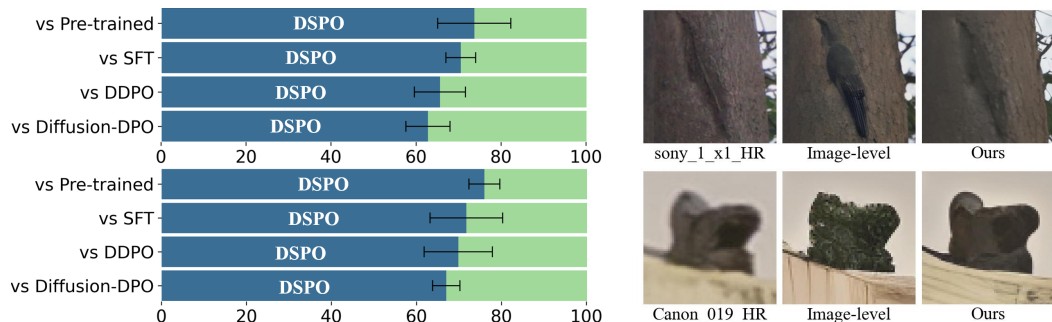

Figure 3: (Left) User preference win rates of DSPO over the pre-trained method, SFT, DDPO, and Diffusion-DPO on DRealSR (top) and RealSR (bottom), based on human annotations. Results include 95% confidence intervals from three independent annotation rounds. (Right) Qualitative comparison of DSPO with image-level preference alignment method.

preference modeling. Furthermore, our empirical results show that the proprietary Qwen2.5-VL-max (with hundreds of billions of parameters) yields more human-aligned scores than our fine-tuned 72B Qwen2.5-VL, while also providing significantly faster inference via remote interface access. Hence, we adopt Qwen2.5-VL-max as our preference scoring model. For the user description feedback strategy, we compute the similarity of the text generated by Qwen2.5-VL-max between the ground truth and the four candidate instance images. When the similarity is less than 0.1, we assume that the instance image is hallucinated, and the corresponding text is selected as the feedback prompt.

**Implementation Details.** During the training and inference of DSPO, we adopt the same hyper-parameter settings as used in the pre-training stage. The DSPO is trained on 8 NVIDIA A100 80GB GPUs, and we set $\beta = 8000$.

We evaluate DSPO against the following preference alignment baseline methods: the pre-trained method (one-step or multi-step SR frameworks), SFT, DDPO Ho et al. (2020), and Diffusion-DPO Wallace et al. (2024). Specifically, the SFT baseline fine-tunes the pre-trained method based only on the subset of images labeled as 'preferred'. Additionally, we compare DSPO with existing Real-ISR methods, including StableSR Wang et al. (2024a), DiffBIR Lin et al. (2024), ResShift Yue et al. (2023), SinSR Wang et al. (2024b), PASD Khan et al. (2023), AddSR Xie et al. (2024), OSED-iff Wu et al. (2025), and SeeSR Wu et al. (2024). We evaluate the test set on both real-world and synthetic data. The real-world data come from RealSR Cai et al. (2019) and DRealSR Wei et al. (2020), containing LQ-HQ pairs at resolutions of $128 \times 128$ and $512 \times 512$. The synthetic set includes 3000 DIV2K-val Agustsson & Timofte (2017) HQ images (512×512), with corresponding LQ images generated using Real-ESRGAN Wang et al. (2021).

**Evaluation of Human Annotator Method** We calculate the user preference win rates (i.e., the frequency with which the human prefers images generated by DSPO) for the human annotator method. We ask annotators to compare images generated by DSPO and another method under the same LQ condition and select the image they prefer (*i.e.*, 'Which image do you prefer given the LQ?').

**Evaluation of Automatic Scoring Method** We utilize a series of metrics to evaluate the results of the automatic scoring method and other methods, including PSNR Wang et al. (2004), SSIM Wang et al. (2004), LPIPS Zhang et al. (2018), DISTS Ding et al. (2020), NIQE Zhang et al. (2015), MUSIQ Ke et al. (2021), MANIQA Yang et al. (2022), and CLIPIQA Wang et al. (2023).

## 6 RESULTS

### 6.1 PREFERENCE ALIGNMENT FROM ANNOTATORS

Fig. 3 (left) shows the user preference win rates of DSPO compared to the pre-trained model and other preference alignment baseline models under the human annotator method, based on the one-

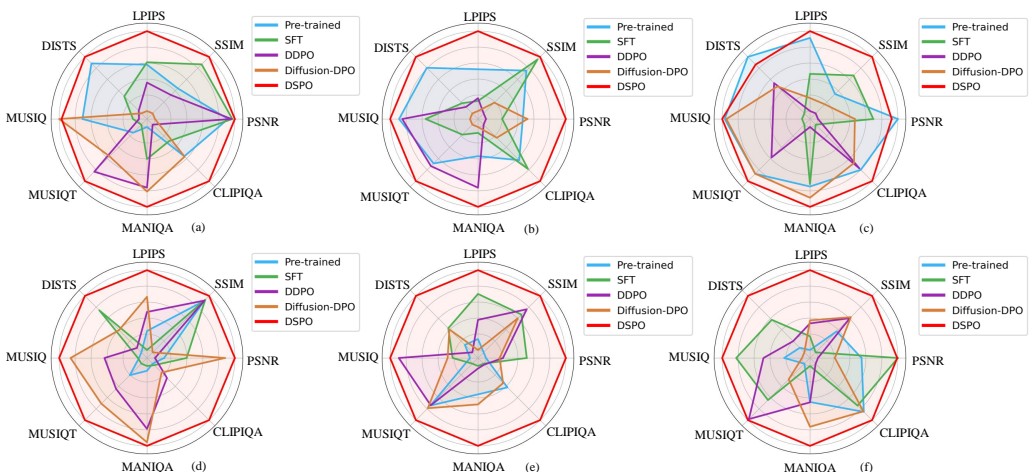

Figure 4: Quantitative comparison of DSPO with the preference alignment baselines on the automatic scoring method. (a)-(c) depict the radar plots for the one-step SR framework on RealSR, DRealSR, and DIV2K-val, while (d)-(f) show radar plots for the multi-step SR framework on the same datasets.

Table 1: Quantitative comparison of DSPO with Real-ISR baseline methods on the DRealSR Dataset. We use bold to emphasize the first, and underline to indicate the second.

| Metrics | StableSR | DiffBIR | ResShift | SinSR | PASD | AddSR | OSEDiff | SeeSR | Ours |
|---|---|---|---|---|---|---|---|---|---|
| PSNR ↑ | 28.03 | 26.71 | 28.46 | 28.36 | 27.36 | 27.77 | 27.92 | 28.17 | **28.54** |
| SSIM ↑ | 0.7536 | 0.6571 | 0.7673 | 0.7515 | 0.7073 | 0.7722 | **0.7835** | 0.7691 | 0.7813 |
| LPIPS ↓ | 0.3284 | 0.4557 | 0.4006 | 0.3665 | 0.3760 | 0.3196 | 0.2968 | 0.3189 | **0.2931** |
| DISTS ↓ | 0.2269 | 0.2748 | 0.2656 | 0.2485 | 0.2531 | 0.2242 | 0.2165 | 0.2315 | **0.2102** |
| NIQE↓ | 6.5239 | 6.3124 | 8.1249 | 6.9907 | **5.5474** | 6.9321 | 6.4902 | 6.3967 | 6.1330 |
| MUSIQ↑ | 58.51 | 61.07 | 50.6 | 55.33 | 64.87 | 60.85 | 64.65 | 64.93 | **66.01** |
| MANIQA↑ | 0.5601 | 0.5930 | 0.4586 | 0.4884 | 0.6169 | 0.5490 | 0.5899 | 0.6042 | **0.6203** |
| CLIPIQA↑ | 0.6356 | 0.6395 | 0.5342 | 0.6383 | 0.6808 | 0.6188 | 0.6963 | 0.6804 | **0.7045** |

step SR framework. Three independent annotation rounds are conducted and the 95% confidence interval of the win rate is provided. It can be observed that DSPO significantly improves the human preference alignment of the pre-trained model, achieving an average win rate of 73.5% and 75.1% on the DRealSR and RealSR, respectively. Furthermore, human annotators prefer the results generated by DSPO over those produced by SFT, DDPO, and Diffusion-DPO. This indicates that for SR-based tasks, the semantic-level DSPO approach is more effective in enhancing alignment with human preferences than image-level human preference alignment methods.

## 6.2 PREFERENCE ALIGNMENT FROM AUTOMATIC SCORING

Fig.4 illustrates the automatic scoring model method compared with the preference alignment baselines for both one-step and multi-step SR models, presented in a radar plot. It can be observed that DSPO consistently outperforms the pre-trained model across all metrics, regardless of one-step or multi-step frameworks. In addition, other image-level alignment methods (SFT, DDPO, Diffusion-DPO) offer inferior gains due to conflicts between image-level preferences and pixel-level objectives in Real-ISR. DSPO alleviates this by introducing instance-level semantic guidance, which better captures semantic structures, reduces sensitivity to local artifacts, and improves overall SR quality. We conduct a visual comparison with the image-level preference alignment method, as exhibited in Fig. 3 (right), with more results provided in the supplementary material. It can be observed that DSPO, through instance-level semantic guidance, avoids hallucinations and artifacts, resulting in a more natural SR image.

As shown in Table 1, we compare our method with existing Real-ISR methods on a general benchmark dataset. Our results are obtained by integrating the proposed DSPO into SeeSR. Through comparisons, DSPO achieves the best or second-best performance on all metrics, thereby fully demon-

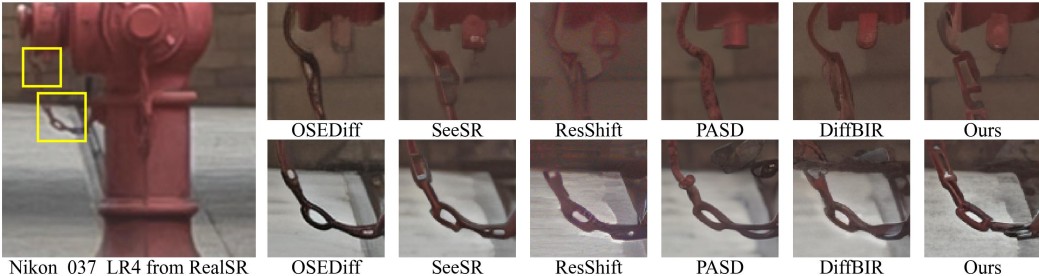

Figure 5: Qualitative comparison of DSPO with Real-ISR baseline methods.

Table 2: Plug-and-Play performance validation of DSPO across multiple SR methods on the DRealSR dataset.

| Method | PSNR↑ | SSIM↑ | LPIPS↓ | DISTS↓ | NIQE↓ | MUSIQ↑ | MANIQA↑ | CLIPIQA↑ |
|---|---|---|---|---|---|---|---|---|
| OSEDiff | 27.92 | 0.7835 | 0.2968 | 0.2165 | 6.4902 | 64.65 | 0.5895 | 0.6963 |
| OSEDiff+DSPO | **27.98** | **0.7919** | **0.2871** | **0.2150** | **6.4411** | **65.80** | **0.6058** | **0.7043** |
| SD2.1 | 27.21 | 0.7598 | 0.3380 | 0.2535 | 6.5110 | 61.16 | 0.5877 | 0.6675 |
| SD2.1+DSPO | **28.08** | **0.7613** | **0.3154** | **0.2357** | **6.3724** | **64.26** | **0.6113** | **0.6819** |
| SeeSR | 28.17 | 0.7691 | 0.3189 | 0.2315 | 6.3967 | 64.93 | 0.6042 | 0.6804 |
| SeeSR+DSPO | **28.54** | **0.7813** | **0.2931** | **0.2102** | **6.1330** | **66.01** | **0.6203** | **0.7045** |
| AddSR | 27.77 | 0.7722 | 0.3196 | 0.2242 | 6.9321 | 60.85 | 0.5490 | 0.6188 |
| AddSR+DSPO | **28.20** | **0.7851** | **0.3056** | **0.2087** | **6.7451** | **62.28** | **0.5600** | **0.6352** |

strating its superior performance. The visualization in Fig. 5 contrasts DSPO with other Real-ISR methods. Image generated from DSPO closely aligns with human visual perception and preserves the edge details and texture of target objects.

In addition, DSPO, as a plug-and-play module, is integrated into four mainstream SR methods. Results in Table 2 show that DSPO significantly enhances key metrics, demonstrating its effectiveness and generalizability as a universal plug-and-play solution.

**Ablation of Different Strategy**   To analyze the effects of the semantic instance alignment strategy and the user description feedback strategy, we conduct an ablation study, as exhibited in Table 3. The experimental results demonstrate that the semantic instance alignment strategy significantly improves preference and reduces artifacts and hallucinations. Incorporating the user description feedback strategy further enhances the image SR performance, leading to more refined results. The combination of both strategies achieves the best overall performance.

Table 3: Ablation study on different strategies. 'M1' represents the semantic instance alignment strategy and 'M2' represents the user description feedback strategy.

| Method | PSNR↑ | SSIM↑ | MANIQA↑ | CLIPIQA↑ |
|---|---|---|---|---|
| Pre-trained | 28.17 | 0.7691 | 0.6042 | 0.6804 |
| M1 | 28.15 | 0.7755 | 0.6123 | 0.6932 |
| M1+M2 | **28.54** | **0.7813** | **0.6203** | **0.7045** |

## 7 CONCLUSION

This paper presents Direct Semantic Preference Optimization (DSPO), a novel framework that pioneers human preference alignment in Real-ISR. To address the dilemma between image-level preference of DPO and pixel-wise preference alignment, our method introduces two key innovations: (1) A semantic instance alignment strategy that optimizes semantic preference learning at the instance level to achieve finer-grained alignment, and (2) a user description feedback strategy that injects user-selected semantic hallucination texts as prompts. Comprehensive experiments demonstrate DSPO's superiority over Real-ISR and preference alignment baselines, and its strong generalizability across both one-step and multi-step frameworks.

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

## A ABLATION ON DIFFERENT OPTIMIZATION OBJECTIVES

To illustrate the impact of different optimization objectives of DSPO on SR, we present visual analyses in Fig. 6. DSPO can be optimized for either perceptual or fidelity. When optimized for perceptual quality, the generated images produce highly realistic stamens, enriching visual details. Conversely, optimizing for fidelity results in images that yield results closer to the GT, accurately preserving structural integrity. These results highlight the adaptability and effectiveness of DSPO in SR tasks.

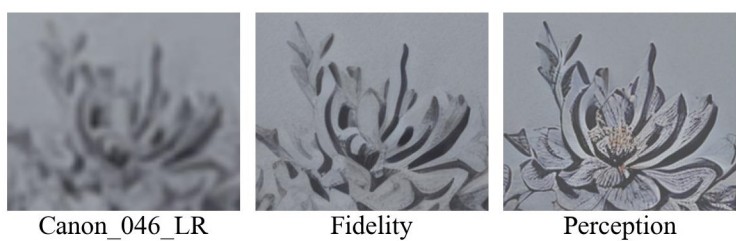

| Canon_046_LR | Fidelity | Perception |

Figure 6: Ablation on different optimization objectives.

## B ABLATION ON $\beta$ IN DSPO LOSS

As shown in Fig.7, an ablation study is presented that examines the impact of various $\beta$ parameter settings of DSPO. The results indicate that setting $\beta$ to 8000 yields an overall better effect, significantly enhancing the effectiveness of the SR task. When $\beta$ is too small, the SR model degenerates into a pure reward scoring model. In contrast, overly large $\beta$ imposes a strong KL-divergence penalty, suppressing any appreciable adaptation.

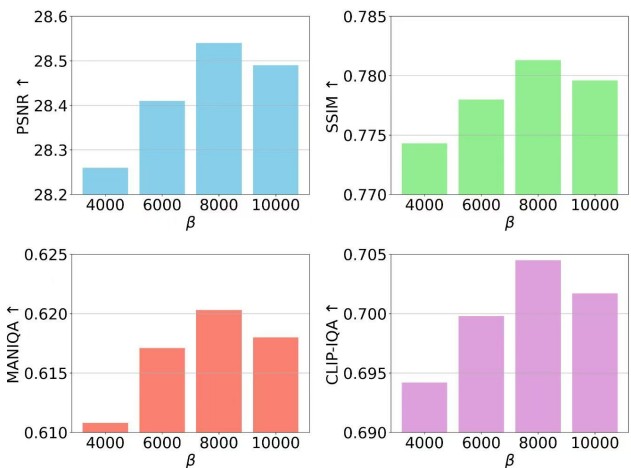

Figure 7: Ablation on $\beta$ in DSPO Loss.

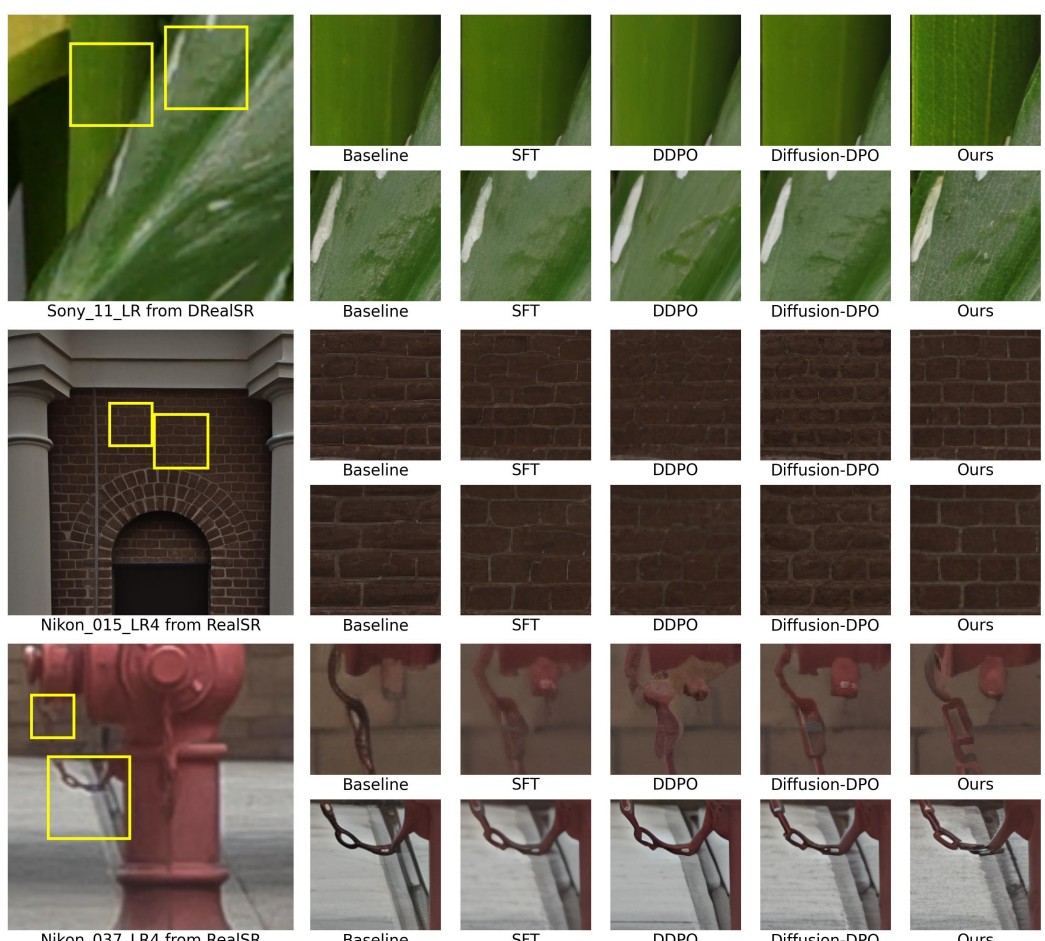

Figure 8: Visualization comparison of DSPO and other preference alignment methods.

## C    VISUAL COMPARISON OF DSPO AND PREFERENCE ALIGNMENT BASELINES

Fig. 8 presents a visual comparison between DSPO and other preference alignment baselines. The results indicate that DSPO generates more realistic textures while effectively suppressing artifacts and blurriness.

## D    REPRODUCIBILITY STATEMENT

We provide detailed descriptions of our model architecture, training procedure, and evaluation metrics in the main text. Additional ablation studies are included in the appendix. We plan to release the source code and pretrained models upon acceptance to facilitate reproducibility of our results.

## E    THE USE OF LARGE LANGUAGE MODELS (LLMS)

We use a large language model (ChatGPT) solely for polishing the language of the paper. It does not contribute to research ideation, experiment design, analysis, or writing of the scientific content.

