# OpenReview forum: "DSPO: Direct Semantic Preference Optimization for Real-World Image Super-Resolution"
_ICLR.cc/2026/Conference — ICLR 2026 Conference Withdrawn Submission_

### Official Review · Reviewer_EtYi · 2025-10-29

**Soundness:** 2
**Presentation:** 2
**Contribution:** 2
**Rating:** 4
**Confidence:** 4

**Summary:**

This paper proposes direct semantic preference optimization to achieve instance-level human preference alignment for real-ISR. In addition, the authors propose injecting hallucination-related semantic text into the prompt to mitigate hallucinations. Experiments across multiple datasets and methods show that the proposed approach improves both fidelity and perception metrics.

**Strengths:**

1. The proposed method, as a plug-and-play approach, can be applied to both single-step and multi-step diffusion models, offering high adaptability.

2. The method achieves significant performance gains on both reference-based and no-reference metrics.

**Weaknesses:**

1. The description of how to select the best and worst instances is very brief and unclear. As the paper’s main innovation, it is recommended to use more text and figures to explain this process. Since SAM may yield a different number of instances for each SR image, how is the correspondence of instances established across different SR results? When selecting the best and worst instances, should one compare the same instance across different images as well as different instances within the same image?

2. The description of the human annotators is insufficiently clear. The procedure used, the criteria, the number of images selected, etc., are all unclear.

3. The description of the automatic scoring is also unclear. What prompts are used to filter instances? What inputs are provided to Qwen2.5-VL-Max—whole images or instances? How is this process concretely manifested as instance-level preference selection?

**Questions:**

1. For one-step models, how is the generation of different SR outputs controlled?

2. Why is the description feedback strategy effective? This strategy is only used during training and is not formulated as a loss—so how is the knowledge from this strategy injected into the model parameters?

---

### Official Review · Reviewer_ahNt · 2025-10-29

**Soundness:** 3
**Presentation:** 1
**Contribution:** 2
**Rating:** 2
**Confidence:** 4

**Summary:**

This paper introduces human preference alignment into Real-World Image Super-Resolution, using Direct Preference Optimization. To address mismatches between pixel-level objectives and image-level preferences, the authors propose Direct Semantic Preference Optimization (DSPO). DSPO combines semantic instance alignment and user description feedback to improve output quality and reduce hallucinations. The method outperforms existing baselines and works well across different SR frameworks.

**Strengths:**

1. Proposes DSPO, effectively combining semantic guidance and user feedback.

2. Reduces artifacts and hallucinations, improving image quality.

3. Plug-and-play method with strong generalizability across SR frameworks.

4. Outperforms existing Real-ISR and preference alignment baselines.

**Weaknesses:**

1. The paper uses four metrics for pre-filtering data, but the motivation is unclear. to select only a portion of the data? Why not rely entirely on Qwen2.5-VL-max? Mixing human annotations with Qwen2.5-VL-max may introduce inconsistencies or bias in defining winners and losers.

2. The number of post-training iterations is not specified, which is important for reproducibility and performance analysis.

3. The paper mentions a "fine-tuned 72B Qwen2.5-VL" model but does not explain what it is or how it was fine-tuned. More details are needed.

4. The concept of instance-level alignment is mentioned but not clearly defined. Additional explanation and ablation studies are needed to show its contribution.

5. The role of the feedback prompt is unclear. Please provide more detailed descriptions and ablation experiments to show its effectiveness.

6. While the paper compares different training methods using performance and user studies, it lacks details about the settings of baseline methods. For example, when comparing DSPO with SFT, DDPO, and Diffusion-DPO, were the same winner and loser candidates used across methods? This should be clarified.

**Questions:**

1. OSEDiff is a one-step diffusion-based SR method. Since it cannot generate multiple solution candidates by adjusting hyper-parameters (e.g., step and cfg), how was data collected for it?

2. Figures 3 and 4 and Table 1 use three different formats (bar chart, radar chart, and table) to compare method performance across datasets. This presentation feels overly fancy. It's recommended to use a consistent format for clarity.

3. Figure 3 compares two datasets (RealSR and DRealSR), while Figure 4 includes three (RealSR, DRealSR, and DIV2K-val). However, Table 1 and Table 2 only compare DRealSR when evaluating other diffusion-based methods. What is the reason for this inconsistency? It is suggested to include more datasets when comparing to diffusion-based baselines to better demonstrate generalization.

---

### Official Review · Reviewer_R2TF · 2025-10-30

**Soundness:** 3
**Presentation:** 2
**Contribution:** 3
**Rating:** 2
**Confidence:** 5

**Summary:**

This paper proposes Direct Semantic Preference Optimization, a new framework for aligning real-world image super-resolution (Real-ISR) models with human preferences. The authors argue that current diffusion-based SR models lack alignment with subjective human judgment, often producing artifacts or hallucinations. To address this, DSPO combines two main strategies:

1. Semantic instance alignment, which breaks down SR outputs into instance-level regionsand applies DPO at that finer granularity.
2. User description feedback, which uses the Qwen2.5-VL-max to generate text descriptions of hallucinated regions and feeds those descriptions back into the model as prompts.

The method is evaluated across multiple SR frameworks, and results show consistent improvements over traditional SR methods and preference alignment baselines like SFT, DDPO, and Diffusion-DPO. DSPO is also designed to be plug-and-play, requiring no changes to the model architecture at inference time.

**Strengths:**

- Good idea. Combining DPO with instance-level semantic segmentation and user feedback is a creative and well-motivated extension of existing preference alignment ideas.

- The experiments are fairly comprehensive, covering multiple SR models and both human and automatic evaluations. The plug-and-play design is practical.

**Weaknesses:**

- **Motivation-method misalignment**: While the motivation that "human preference may differ across regions of the image" is compelling, the method for identifying such regions—using SAM—is problematic. SAM is not tied to human preference in any way; it segments based on low-level image cues and may completely miss the regions that actually drive human judgments (e.g., faces, text, artifacts). Worse, if the selected regions do not contain the key preference-differentiating content, the model receives little to no useful optimization signal. In such cases, full-image preference comparison may even be more effective due to its coverage. This introduces a critical gap between the intended motivation (fine-grained preference alignment) and the actual implementation.

- **Inconsistent regional alignment**: Moreover, the paper applies SAM separately to each SR output image, meaning that the segmented regions differ across outputs. This makes it unclear how a fair instance-level preference comparison can be made across different SR images, as the masks refer to potentially non-overlapping content. The paper does not address how it aligns or matches these regions across images, which is a critical detail for any instance-level optimization.

- **Limited diversity in preference sampling**: The method generates only 4 SR outputs per LQ image by varying inference parameters (CFG and steps) of a single model. This narrow sample pool may lack sufficient diversity to produce meaningful or representative human preferences. If all samples are too similar or none are particularly good, the resulting preference signal could be weak or noisy. This design potentially limits the effectiveness of the instance-level DPO training.

-  **Inappropriate FR IQA metrics** The paper combines full-reference and no-reference IQA metrics to select SR candidates, which is a good idea in principle. However, the choice of PSNR and SSIM feels outdated and somewhat inconsistent with modern SR evaluation practices—especially for real-world SR task. PSNR is known to favor overly smooth results, and SSIM, while better, still struggles to reflect perceptual quality. More perceptually aligned metrics like LPIPS or DISTS would have been more appropriate as full-reference metrics here.

- **Unclear sample generation for OSEDiff**: The DSPO framework relies on generating multiple SR outputs per image by varying inference parameters like CFG or sampling steps. However, OSEDiff is a one-step model that does not support such parameters. The paper does not explain how multiple samples are obtained for OSEDiff. This raises concerns about the consistency and fairness of training or evaluation across models.

- **Limited baseline diversity**: All experiments in the paper are conducted on the SD2 series of base models. While this ensures consistency, it also limits the generality of the conclusions. More recent and stronger diffusion backbones such as SDXL (e.g., FaithDiff, CVPR 2025) and SD3 (e.g., DIT4SR for multi-step and TSDSR for one-step, both from ICCV/CVPR 2025) have shown significant improvements in SR quality. Evaluating DSPO on these stronger baselines would provide a more comprehensive understanding of its effectiveness and robustness across architectures.

- **Citation error [1]**: The paper repeatedly refers to "DDPO" and attributes it to Ho et al. (2020), which is actually the original DDPM paper. The correct reference for DDPO—i.e., applying reinforcement learning to diffusion models—is likely "Training diffusion models with reinforcement learning" by Kevin Black et al. (2023). This misattribution creates confusion and should be corrected, especially since DDPO is a key baseline in the paper.

- **Citation error [2]**: The paper attributes PSNR to Wang et al. (2004), which is incorrect. That paper introduced SSIM, not PSNR. PSNR is a much older, standard signal processing metric and should not be credited to Wang et al. This isn’t a major issue, but it’s worth correcting for accuracy. Line 301

**Questions:**

Please see the weaknesses.

---

### Official Review · Reviewer_Edvj · 2025-10-31

**Soundness:** 2
**Presentation:** 2
**Contribution:** 3
**Rating:** 2
**Confidence:** 5

**Summary:**

This paper pioneers the application of human preference alignment to real-world image super-resolution (Real-ISR). It aims to resolve the misalignment between the outputs of powerful diffusion-based SR models and human perceptual preferences.
1. To address this, the paper introduces Direct Semantic Preference Optimization (DSPO), a framework featuring two core innovations:
Semantic Instance Alignment: The preference learning is performed at the semantic instance level (e.g., objects, regions) rather than on the entire image.
2. User Description Feedback: The framework leverages a Vision-Language Model (VLM) to generate textual descriptions of generated instances. Descriptions corresponding to hallucinations or artifacts are then used as negative prompts to actively suppress such failures.

**Strengths:**

1.  This is the first work to systematically introduce advanced preference alignment techniques like DPO into the image SR domain. It successfully bridges the fields of AIGC Alignment and Low-level Vision, opening a promising new direction for improving the quality and controllability of SR models.
2.  The paper convincingly demonstrates the capability of DSPO by applying it to four different one-step and multi-step SR frameworks (OSEDiff, SD2.1, SeeSR, AddSR), showing consistent improvements across the board (Table 2).
3. The paper is exceptionally well-written and logically structured. The motivation is clearly articulated, and the high-quality figures (especially Fig. 1 and 2) are highly effective at illustrating the core problem and the proposed method.

**Weaknesses:**

1. The automatic preference labeling pipeline relies heavily on a powerful, proprietary VLM (Qwen2.5-VL-max). This raises concerns about reproducibility and performance attribution. It is unclear how much of the performance gain comes from the DSPO framework itself versus the near-perfect judgments of the VLM. The performance with weaker, open-source VLMs is not discussed.
2. The instance-level approach is clever, but its handling of conflicts seems overly simplistic. For a complex scene, one candidate image might excel on a "sky" instance while another excels on a "building" instance. The current method of selecting the entire image based on one winning instance may discard equally good content from other candidates.
3. The user feedback strategy relies on injecting negative text prompts. While this is straightforward for text-conditional SR models like SeeSR, it is unclear how this mechanism would be integrated into the many SR models that are not text-conditional.

**Questions:**

1. How sensitive is DSPO's performance to the quality of the VLM used for automatic labeling? Have you experimented with smaller, publicly available VLMs (e.g., LLaVA) to assess this?
2. Could you clarify how the global preferred (xw) and dispreferred (xl) images are selected when the best instances originate from different candidate images? The paper states, "The whole image xw of the best instance xm is defined as the preferred example," which implies that high-quality instances in other images might be unfairly penalized.
3. For the user description feedback, how are negative prompts incorporated into SR models that are not text-conditional? Does this imply the strategy is only applicable to a subset of SR architectures?
4. The human study involved 10 professionals. Could you provide an estimate of the total time and effort required for the annotation process?
5. DPO fine-tuning in LLMs often converges quickly. Could you provide a convergence analysis (e.g., loss curves) for the DSPO fine-tuning stage to demonstrate its training stability and efficiency?
6. Why were more recent and powerful SR models, such as SUPIR, not considered for validation?

---

### Note · Authors · 2025-11-13

I have read and agree with the venue's withdrawal policy on behalf of myself and my co-authors.